# Technical Approaches to the Recycling of Reclaimed Asphalt Pavement into Aggregate and Binder

**Kengo Akatsu, Yousuke Kanou * and Shouichi Akiba**

College of Industrial Technology, Nihon University, 1-2-1 Izumi-Cho, Narashino 275-8575, Japan;
akatsu.kengo@nihon-u.ac.jp (K.A.); akiba.shouichi@nihon-u.ac.jp (S.A.)
* Correspondence: kanou.yousuke@nihon-u.ac.jp

**Abstract:** Approximately 99% of reclaimed asphalt pavement (RAP) has been recycled in Japan in recent years. However, the deterioration in quality of repeatedly recycled RAP cannot be prevented through existing methods, nor can sustainability be guaranteed. In addition, it is challenging to procure virgin aggregate and binder. Therefore, to ensure the quality and supply of future recycled hot-mix asphalt, it is necessary to explore sustainable recycling technologies. This study examined the advantages and disadvantages of the technical approaches to the recycling of RAP into aggregate and binder. We develop a recycling technology (separate recycling technologies) that uses hot water to separate and restore aggregate and binders to their initial condition from RAPs. The quality of the aggregate, recovered by the hot water rubbing method at 80 °C and 90 °C, fully satisfies the standard values for virgin aggregate at all temperatures. The aged binders, reacting through a hydrothermal decomposition method (hydropyrolysis), with a reaction temperature of 300–350 °C and a reaction time of 0–15 min, tend to have a significantly improved effect. These results confirm that both the hydrothermal rubbing and hydropyrolysis methods could be beneficial options for establishing separate recycling technologies for RAP.

**Keywords:** reclaimed asphalt pavement; recycling; hydropyrolysis; hydrothermal rubbing; binder; aggregate

## 1. Introduction

Recycling technology for reclaimed asphalt pavements (RAPs) in Japan was first developed in the 1970s and has been commercialized since the 1980s [1]. In recent years, approximately 99% of RAP has been recycled, and recycled hot-mix asphalt (RHMA) accounts for approximately 75% of the total shipment of asphalt mixtures [2]. Simultaneously, with the widespread adoption of direct heating plants with twin dryers, the RAP content has already exceeded 50% in the national average of RHMA shipments.

As shown in Figure 1, RHMAs recycled for a third time will shortly be in circulation, according to estimates based on RHMA shipments and RAP content history. Therefore, experience and knowledge have been gathered to sustain the product quality of RHMA. However, the quality of the old asphalt binder in RAP has deteriorated due to repeated reclamation and may degrade the quality of RHMA in the future. In addition, existing technology does not prevent quality degradation and ensure sustainability of the old asphalt binder in RAP. Kawakami et al. [3], in their studies of the repeated regeneration of asphalt, have shown that elongation is not restored. Furthermore, various modified binders have become widespread in Japan since the 1990s, and the quality of RAP has diversified. Therefore, multifaceted evaluation methods for the performance of repeatedly recycled RHMA are being investigated, along with rejuvenating agents.

From a material supply perspective, RAPs will become an important source of material, as the procuring of virgin aggregate and binder is challenging due to the necessity to reduce mining and $CO_2$ emissions. The supply of binders is especially at risk with the drive for

independence from fossil fuels. Therefore, to ensure the quality and supply of RHMA in the future, we must continue to explore sustainable recycling technologies while improving the continuity of recycling using existing technology.

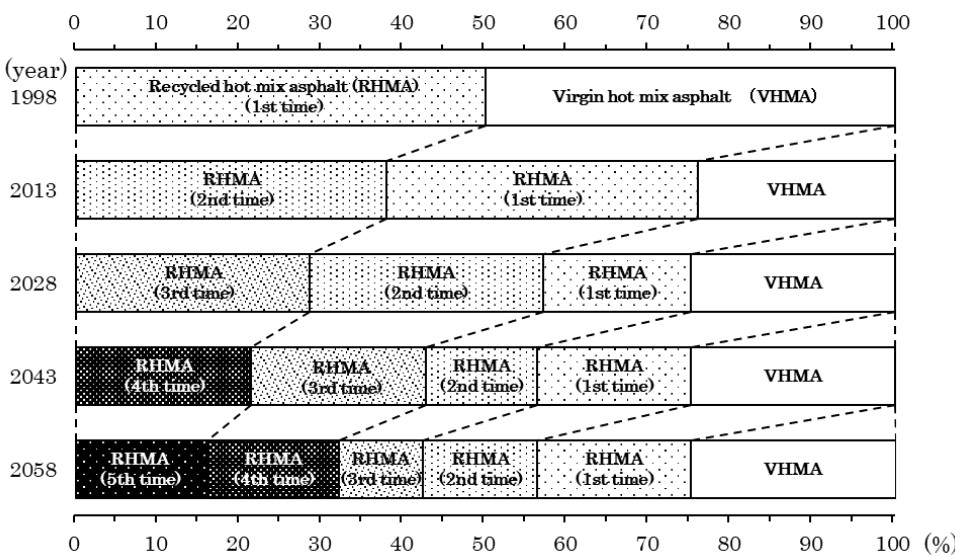

**Figure 1.** Chronology of recycled hot-mix asphalt (RHMA) shipments and reclaimed asphalt pavement (RAP) content.

Several methods for separating RAP into aggregates and binders have been investigated, with a focus on the sustainable recycling of aggregates. We are in the process of developing a separate recycling technology that uses inexpensive and easy-to-handle hot tap water to separate and recover old aggregates and binders that constitute RAPs and restore them to their initial condition [4,5]. This novel technology aims to sustainably recycle aggregates and binders in RAPs; the initial process (the hydrothermal rubbing method) separates the aggregate and binder, and thereafter the binder is rejuvenated (hydropyrolysis). The only resources used for these methods are water and heat. This study reviews the technical approaches for the separate recycling of RAP into aggregate and binder, and thereafter reports the results obtained from the separate recycling technology by distinguishing between the hydrothermal rubbing [6] and hydropyrolysis methods [7].

## 2. Overview of Separate Recycling

Because Japan, is deficient in aggregate resources, recycling technology for separating and recovering aggregates from construction waste, mainly for concrete, has been studied since the 1970s [8]. The rubbing method, in which the lumps collide with each other, has primarily been adopted in practice for separating aggregates from concrete waste. Conversely, the investigation of separating aggregates from asphalt concrete waste only started in the 1990s, based on the precedent of concrete recycling. Studies have also been conducted to prevent the binder from simultaneously reattaching to the separated aggregates. In this section, the method to separate the aggregates is discussed for each target base material, concrete, and asphalt concrete.

### 2.1. Separating Aggregate from Concrete

In 1996, Sugiyama et al. [9] evaluated the effects of the concrete waste crushing method on the quality of recycled aggregates. They concluded that it was difficult to achieve quality standards even with changes to the type and number of crushers commercially available.

In 1999, Yanagibashi et al. [10] attempted to manufacture high-quality recycled aggregates using an eccentric rotor-type recycled coarse aggregate manufacturing device. This method separated the aggregate and mortar by promoting the rubbing action of the concrete waste. As a result, the authors disclosed in 2003 [11] that this method could

be applied to separate and recover high-quality coarse aggregates. However, it did not sufficiently correspond to the recovery of fine aggregates.

In 2000, Shima et al. [12,13] attempted to recover higher-quality aggregates using the heat-rubbing method, which focused on the heat-sensitive characteristics of hardened cement. In this method, concrete waste was heated and agitated with a crushing medium such as iron balls to promote the rubbing effect and separate the aggregate and mortar. The study found that the aggregate separated using this method had the same quality as virgin aggregate. However, because it required heating, there were concerns that complex equipment at a high cost would be required.

In 2004, Yoda et al. [14] proposed a practical mechanical rubbing method for the quality, processing capacity, and cost of aggregate recovered from concrete waste. In this method, the drum-type device was separated by a partition plate, which rotated to promote the rubbing effect of the aggregate and iron balls, thereby separating the aggregate and mortar. The method experienced problems with the fine aggregate quality; however, it efficiently separated and recovered aggregates from concrete waste.

### 2.2. Separating Aggregate from Asphalt Concrete

In 1999, Hisari et al. [15] developed a technology for separating asphalt and aggregate using an oil-absorbing powder such as waste cement concrete powder. In this method, asphalt was absorbed by the powder by heating and mixing the powder and RAP, thus separating the aggregate. The study results indicated that both coarse and fine aggregates separated from RAP containing various modifiers could be reused in asphalt mixtures. However, the condition and utilization of asphalt-absorbed powders as pavement materials have become a new issue.

In 2008, Yamada [16] proposed a recycling technology that applied the grinding method for separating aggregates from RAP containing modified asphalt, which had been a problem for recycling. In this method, the binder film around the coarse aggregate was separated through rubbing of the RAP with the metal partitions or iron balls in the grinder. The resultant coarse aggregates could be reused in the asphalt mixture. However, the problem of recycling fine aggregates and fine particles containing modified asphalt remained unresolved. Neither method described here resolved the quality deficiency of aged asphalt.

We focused on the solvent performance of water for asphalt mixtures and studied its application to asphalt mixture peeling tests in 2013 [17], and to asphalt extraction tests in 2016 [18]. Through the separation process of aggregate and asphalt, and taking advantage of the moisture damage and the decomposition process of asphalt with high-temperature and high-pressure water, the possibility was established that the aggregate and binder could be efficiently separated by rubbing the asphalt mixture in hot water and rejuvenating the aged binder by a subcritical water reaction. Therefore, separate recycling technologies were developed from a series of processes, as shown in Figure 2. These technologies include the hydrothermal rubbing method (Section 3.1), which is the first process, and hydropyrolysis (Section 3.2), which is the second process.

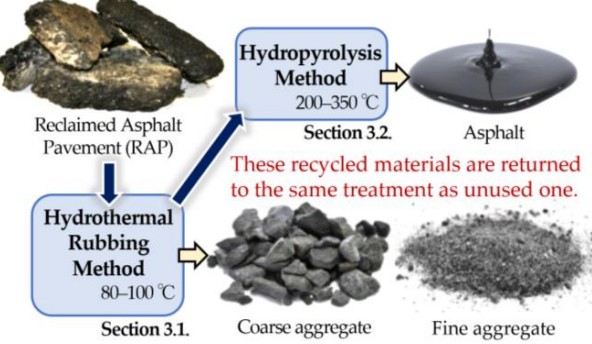

**Figure 2.** Separate recycling technology.

### 3. Methods

*3.1. Hydrothermal Rubbing Method*

In the hydrothermal rubbing method, the RAP is agitated in hot water at normal pressure to break it into granules, and the binder is separated to recover the aggregate. The rubbing in hot water suppresses the reattachment of the binder to the surface of the aggregates.

The procedure for the laboratory experiments is shown in Figure 3. The experiment consisted of the RAP heating process; the first process separated and recovered aggregates $\geq$ 5 mm, and the second process separated and recovered passed material $\geq$ 1 mm. An outline of the equipment used in each process is shown in Figures 4 and 5.

**Heating and Curing:** Water curing of specimens

Cure the specimens (22 kg worth of specimens for wheel-traking) in hot water (40 L) that has reached the test temperature. Its specimens which was curing for 20 min and hot water are transfer to first separating device.
※ Incidentally, specimens is divided into 4 equal parts according to the size of the separating device.

**First Process:** Separating and recovering to 5–13 mm aggregate (SR5–13)

1. While keeping temperature of the hot water, the specimens are crushed and classified by agitating the stirring blade in the device is stirred at 50 rpm and 150 rpm for 5 minutes, respectively.
2. The stirring blade is agitated at 100 rpm for 25 min and the old asphalt is sorted out from the aggregates larger than 5 mm while classifying with a 5mm sieve.
3. While stirring at 100 rpm, SR5–13 is recovered after the 5 mm passing material and hot water are discharged.

Assessment the properties and quality of the Recovered Aggregate (SR5-13)

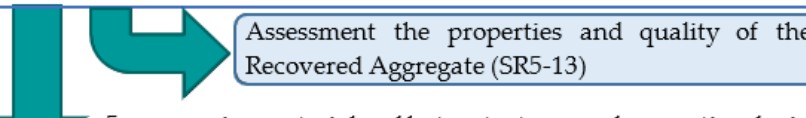

5 mm passing material and hot water to second separating device

**Second process:** Separating and recovering to 1–5 mm aggregate (SR1–5)

1. The stirring blade is agitated at 400 rpm for 25min and 150rpm for 5 min, and the old asphalt is sorted out from the aggregates larger than 1 mm while classifying with a 1mm sieve.
2. While stirring at 400 rpm, SR1–5 is recovered after the 1 mm passing material(SR0–1) and hot water are discharged.
※ Incidentally, separating between SR1–5 and SR0–1 is done in order to collect old asphalt in particle of aggregate that has little influence on recycled aggregate and recover 1–13 mm aggregate.

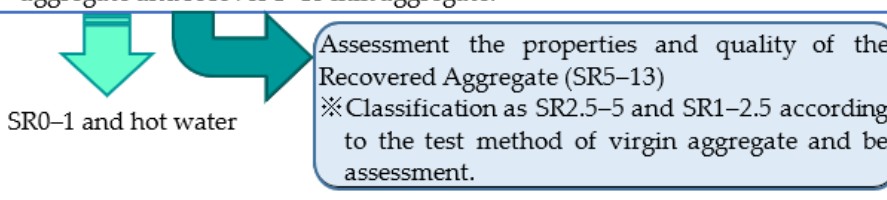

SR0–1 and hot water

Assessment the properties and quality of the Recovered Aggregate (SR5–13)
※ Classification as SR2.5–5 and SR1–2.5 according to the test method of virgin aggregate and be assessment.

**Figure 3.** Experimental flow adopted hydrothermal rubbing method.

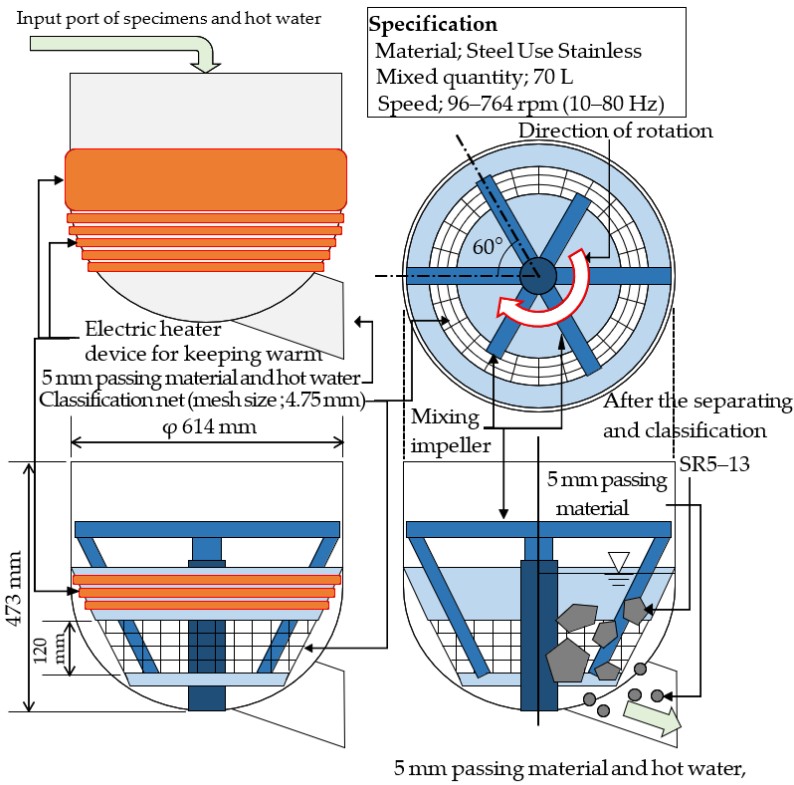

**Figure 4.** First separating device.

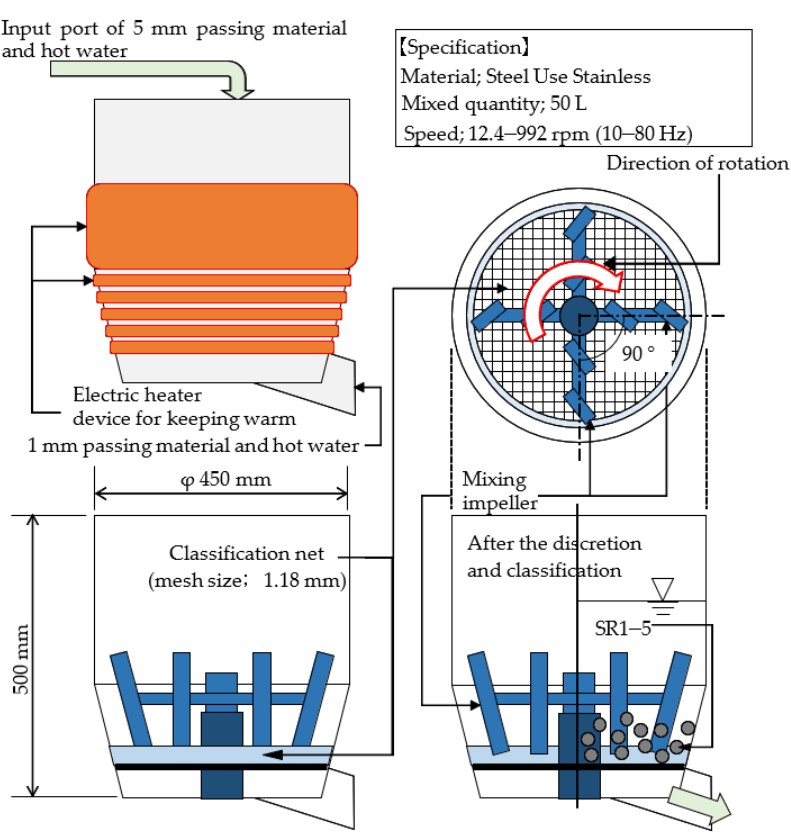

**Figure 5.** Second separating device.

Table 1 lists the compositions of the specimens. Two specimens were used for the wheel-tracking test, in which 5–13 mm RAP (R5–13) and 0–5 mm RAP (R0–5) were mixed with a commercial asphalt mixture. These were compacted to meet standard values, using the density and void ratios as indicators. Table 2 lists the properties of R5–13 and R0–5. The RAP of common quality in Japan is used for relative comparison of the influence of aggregate quality before and after the test.

**Table 1.** Compositions of the specimens.

| Composition | Unit | R5–13 | R0–5 |
|---|---|---|---|
| Compounding rate | % | 46.7 | 53.3 |

**Table 2.** Properties of R5–13 and R0–5.

| Properties | | Unit | R0–13 | R5–13 | R0–5 |
|---|---|---|---|---|---|
| Old asphalt penetration | | 1/10 mm | 21 | 21 | 20 |
| Old asphalt content | | % | 5.9 | 4.0 | 7.7 |
| Percentage passing by mass | 13.2 mm | % | 100.0 | 100.0 | 100.0 |
| | 4.75 mm | | 67.3 | 33.2 | 100.0 |
| | 2.36 mm | | 49.8 | 22.8 | 75.7 |
| | 1.18 mm | | 40.9 | 19.8 | 62.2 |
| | 0.6 mm | | 32.3 | 16.3 | 47.7 |
| | 0.3 mm | | 22.8 | 12.1 | 33.0 |
| | 0.15 mm | | 14.0 | 9.5 | 18.3 |
| | 0.075 mm | | 6.9 | 5.5 | 8.2 |

Separation temperatures of 70 °C, 80 °C, and 90 °C were selected based on the coarse aggregate peel resistance test to facilitate peeling in each process. The rubbing speed was set based on preliminary experiments, taking the binder separation and aggregate wear into account. The 5–13 mm aggregate (SR5–13) and the 1–5 mm aggregate (SR1–5) separated and recovered by this method were left for 1 h until the temperature became constant, and thereafter their properties and quality were evaluated.

*3.2. Hydropyrolysis*

The state of water changes, as shown in Figure 6, owing to the balance between the intramolecular interaction related to temperature, pressure, and kinetic energy. When it exceeds a critical point, which is the endpoint of the gas liquid coexistence line, it becomes a fluid without a gas–liquid interface and exhibits completely different characteristics to water at room temperature (10–30 °C). The solvent characteristics of high-temperature and high-pressure water can be approximately understood from the behavior shown in Figure 7. The temperature changes and the relative permittivity are related to the polarity and ion product, which indicate the hydrolysis performance.

It is recognized that few organic substances with low polarity dissolve in ordinary water, indicative of the so-called relationship between water and oil that do not mix. However, as shown in Figure 7, the relative permittivity, which indicates the polarity, becomes a value comparable to that of an organic solvent as it approaches the critical point. The ionic product of water is usually approximately $1.0 \times 10^{-14}$; however, in the subcritical region, it increases to approximately $1.0 \times 10^{-11}$, which dramatically increases the hydrolysis performance.

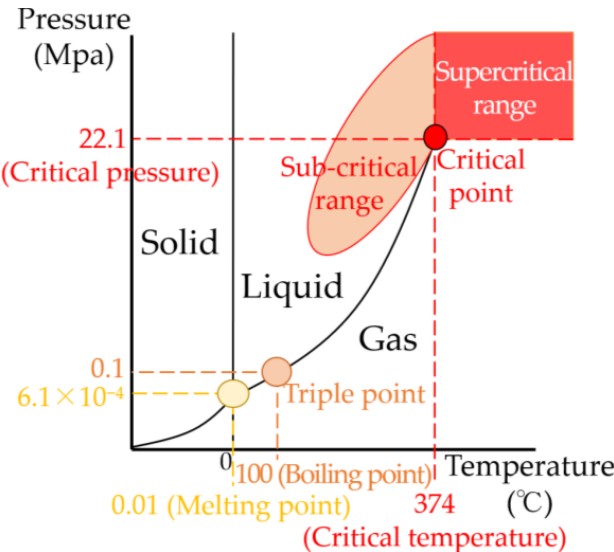

**Figure 6.** Pressure–temperature (P–T) diagram of water.

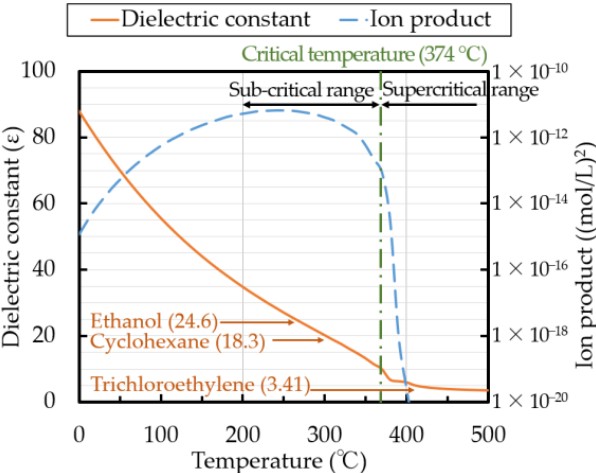

**Figure 7.** Dielectric constant and product.

The authors attempted to rejuvenate the binder using the hydrothermal decomposition method, focusing on the solvent performance of high-temperature and high-pressure water at 200–350 °C, which had already been used in many practical cases [19]. Table 3 shows the outline of the equipment used in the hydropyrolysis laboratory experiment, and Figure 8 shows the procedure. In the experiment, 50 g of the aged binder and the quantity of saturated water reached after the reaction temperature and vapor pressure were reached were charged and heated in a 1000-mL airtight container. After heating for 15 min at each reaction temperature, a sample was collected. The authors evaluated the effects of the reaction temperature, reaction time, and cooling method on the rejuvenation effect. The effects of the reaction temperature at 200 °C, 250 °C, 300 °C, and 350 °C are described in detail in the next section as a representative of the results. The sample used was an accelerated aging asphalt binder (AGI), which was an original asphalt binder (ORG) with a penetration of 67 (1/10 mm) accelerated to a penetration of 20 (1/10 mm). To achieve the prescribed penetration of AGI, the ORG was subjected to thin-film heating of ORG for 5 h (163 °C), and a pressurized aging vessel, a US SHRP test method, was applied to test for the accelerated aging of asphalt binder reaction for 23 h (100 °C, 2.1 MPa). The rejuvenating effect of hydrothermal decomposition on AGI was evaluated physically and chemically. Table 4 shows the properties of ORG and AGI.

**Table 3.** Specifications of equipment.

| | |
|---|---|
| Volume | 1000 mL |
| Allowable pressure | 19 MPa |
| Allowable temperature | 360 °C |
| Container material | SUS 316 |
| Heater method | Aluminum block heater |
| Applicable law | Small pressure vessel |

Appearance of Equipment

**Heating process**  Temperature: 200, 250, 300, 350 °C
Place water and AGI (50 g) in a sealed container and heat for a certain period of time until test temperature and pressure are reached.

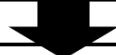

**Reaction process**  Reaction time: 15 min
Let the reaction time be after the test temperature and pressure have been reached and subject the AGI to a high-temperature and high-pressure water reaction.

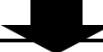

**Cooling process**
In order to suppress degradation during cooling, exhaust pressure cooling is performed until the inside reaches 100℃ and then water cooling is performed.

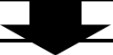

**Dehydration process**  Temperature: 250 °C, Reaction time: 15 min
After cooling to 40 °C, the asphalt is recovered, reacted with superheated steam, and dehydrated.

**Figure 8.** Experimental procedure.

**Table 4.** Properties of ORG and AGI.

| Properties | Unit | ORG | AGI |
|---|---|---|---|
| Penetration | 1/10 mm | 67 | 20 |
| Softening point | °C | 47.3 | 64.9 |
| Elongation | cm | 100+ | 6 |

## 4. Results and Discussion

### 4.1. Hydrothermal Rubbing Method

4.1.1. Properties of Recovered Aggregate

The properties of the recovered aggregate in the rubbing method were evaluated based on the water content, aging binder content, fine particle fraction, and particle size curve. The appearances of SR5–13 (80 °C) and SR1–5 (80 °C) are shown in Figures 9 and 10, respectively, and their properties are listed in Tables 5 and 6, respectively. The appearance of

SR5–13 was comparable to that of a new aggregate, with only a small amount of aged binder remaining on the surface; there was no noticeable wear or damage. The water content and aging binder content tended to decrease as the separation temperature increased. However, the quantity of fine particles was approximately the same at each temperature.

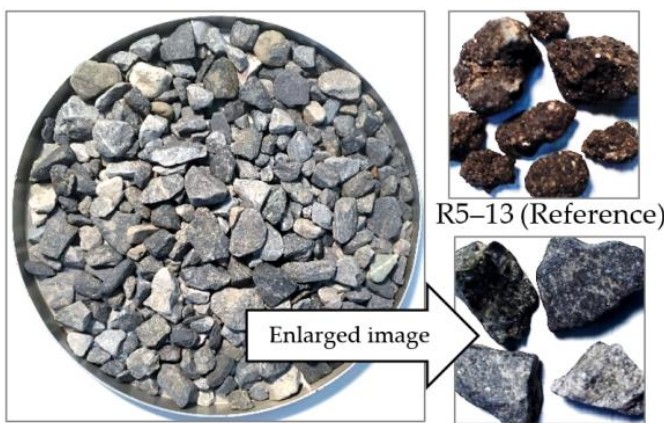

**Figure 9.** Appearance of SR5–13 (80 °C).

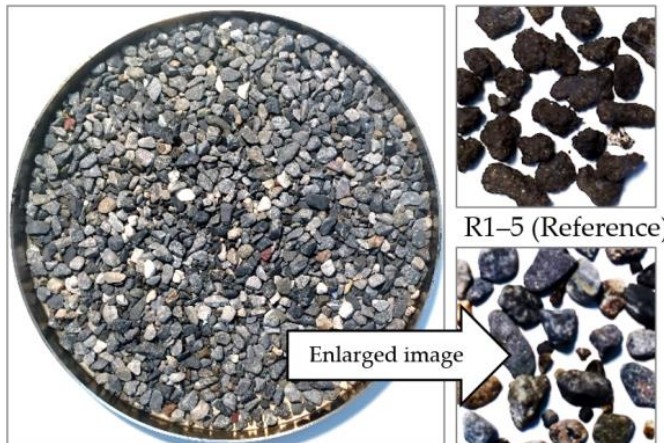

**Figure 10.** Appearance of SR1–5 (80 °C).

**Table 5.** Properties of SR5–13.

| Properties | Unit | Separation Temperature (°C) | | |
|---|---|---|---|---|
| | | **70** | **80** | **90** |
| Moisture content (After natural cooling) | % | 0.83 | 0.77 | 0.57 |
| Old asphalt content | % | 0.78 | 0.70 | 0.56 |
| Content of aggregates finer than 75 μm sieve | % | 0.2 | 0.2 | 0.2 |

**Table 6.** Properties of SR1–5.

| Properties | Unit | Separation Temperature (°C) | | |
|---|---|---|---|---|
| | | **70** | **80** | **90** |
| Moisture content (After natural cooling) | % | 1.47 | 1.08 | 1.04 |
| Old asphalt content | % | 0.68 | 0.57 | 0.44 |
| Content of aggregates finer than 75-μm sieve | % | 0.1 | 0.2 | 0.2 |

The particle size curves are shown in Figures 11 and 12. For reference, the particle size after extracting the binder from the recycled aggregate using the current method is

also described as follows. When the separating temperature is 70 °C, SR5–13 contained a small quantity of aggregate < 4.75 mm; however, it was strictly classified compared to R5–13 evaluated by the current method. For SR1–5, the particle size curves were almost the same for each separation temperature, and there was no difference in the particle size after extracting the binder from them.

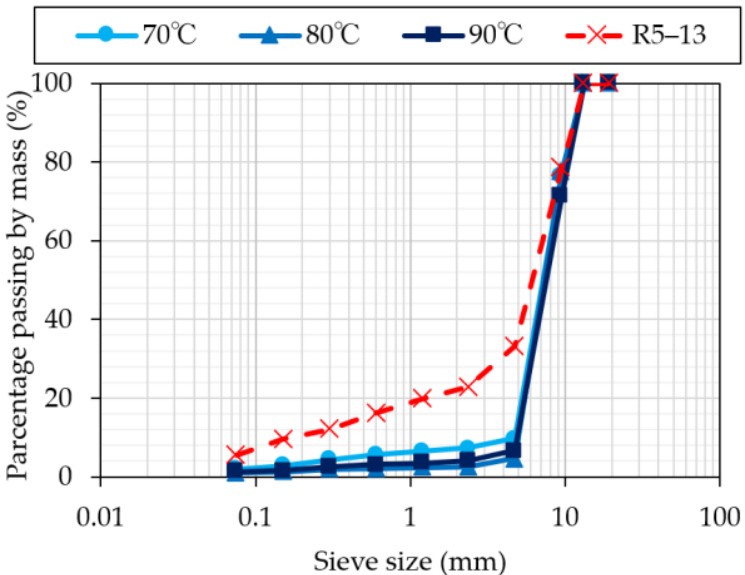

**Figure 11.** Granularity of SR13–5 (after extraction).

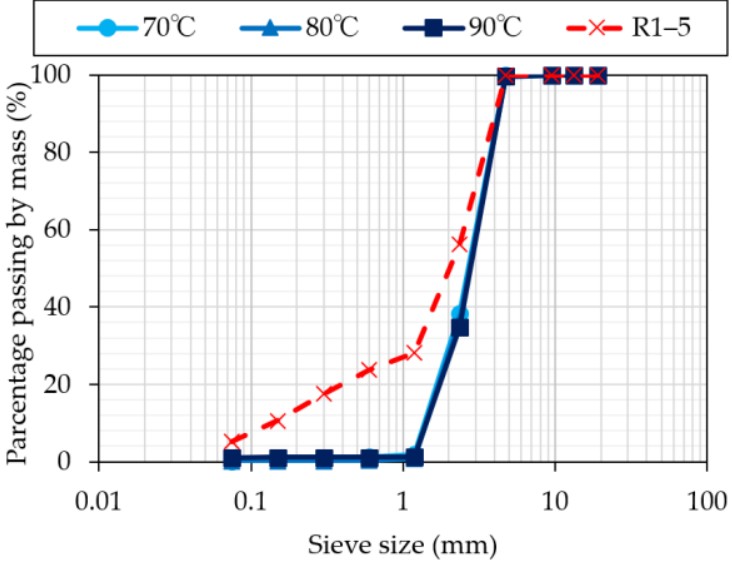

**Figure 12.** Granularity of SR5–1 (after extraction).

4.1.2. Quality of Recovered Aggregate

The quality of the recovered aggregate in the rubbing method was evaluated based on the density, water absorption rate, and wear-down weight, as in the case of virgin aggregate. The density, water absorption rate, and wear loss of SR5–13 are shown in Figure 13, and those of SR1–5 are shown in Figures 14 and 15, respectively. SR1–5 was measured separately according to the respective test methods for coarse aggregate (2.5–5 mm) and fine aggregate (1–2.5 mm). After binder extraction, each aggregate was used as a comparative sample for these evaluations.

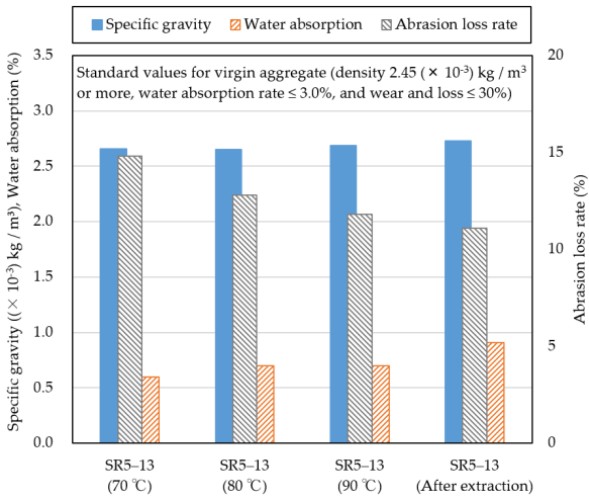

**Figure 13.** Quality of SR5–13.

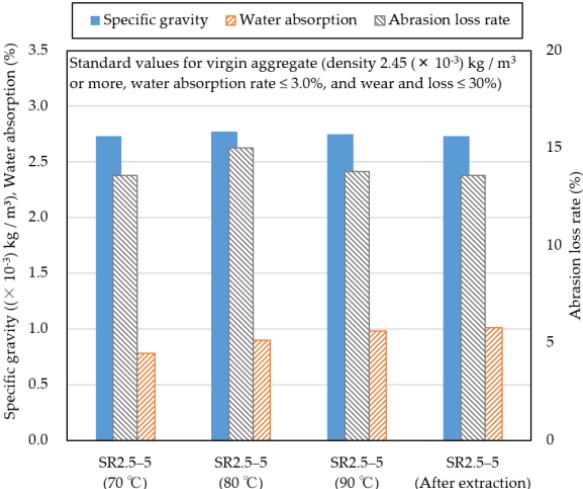

**Figure 14.** Quality of SR2.5–5.

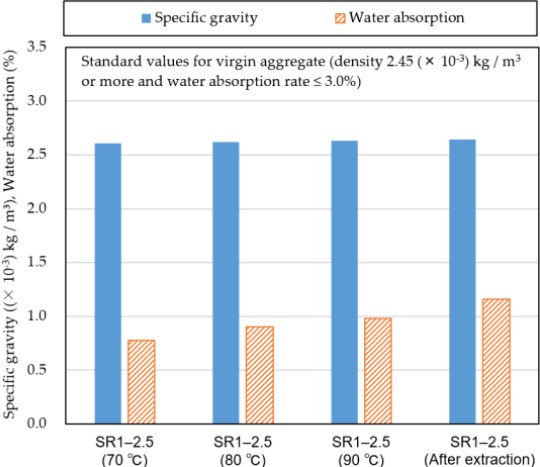

**Figure 15.** Quality of SR1–2.5.

The density of SR5–13 did not differ before and after asphalt extraction; however, the water absorption rate increased slightly as the separation temperature increased. This was because a part of the aging binder was impregnated inside the aggregate; however, it did not affect the quality control and reuse. The wear loss of SR5–13 decreased as the separation temperature increased. The difference before and after asphalt extraction

was considered to be due to the wear or exfoliation of the aged binder remaining in the recovered aggregate. Although the densities of SR2.5–5 and SR1–2.5 were identical before and after binder extraction, the water absorption rate tended to increase as the separation temperature increased, similar to SR5–13. The wear loss of SR2.5–5 did not change before and after binder extraction, and there was no difference due to the separation temperature.

*4.2. Hydropyrolysis*

4.2.1. Chemical Properties of Rejuvenated Binder

Figure 16 shows the changes in molecular weight distribution contingent on the reaction temperature. The rejuvenated binder legend in each chart is shown in parentheses, with the reaction time (e.g., 350(15) for 15 min at 350 °C). Oxidative polymerization and polycondensation, associated with aging, appear as shoulders on the polymer side in AGI. Conversely, the binder rejuvenated by hydrothermal decomposition tended to have smaller shoulders, and the molecular weight distribution of the binder at a reaction temperature of 350 °C approached the same normal distribution as ORG.

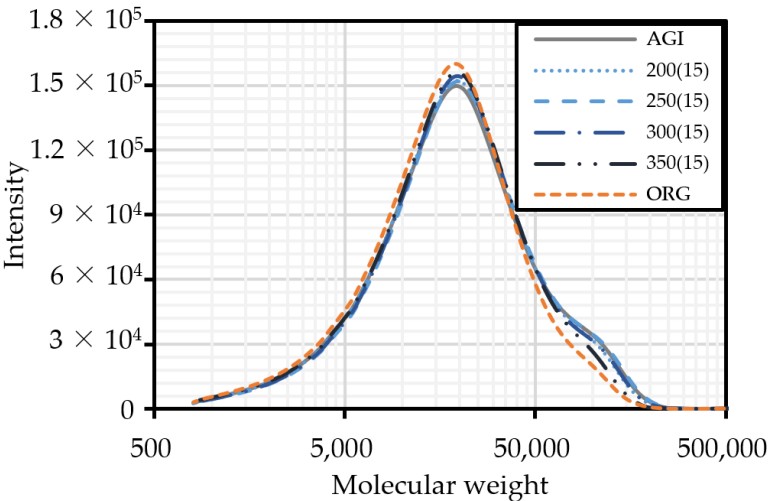

**Figure 16.** Molecular weight distribution (200–350 °C, 15 min).

Figure 17 shows the change in the carbonyl index (CI) with the reaction temperature. The CI of AGI was more significant than that of the ORG, and an increase in the carbonyl groups was observed with aging. Conversely, the binder rejuvenated by hydrothermal decomposition tended to decrease the CI, and the increase due to aging decreased by half at a reaction temperature of 350 °C.

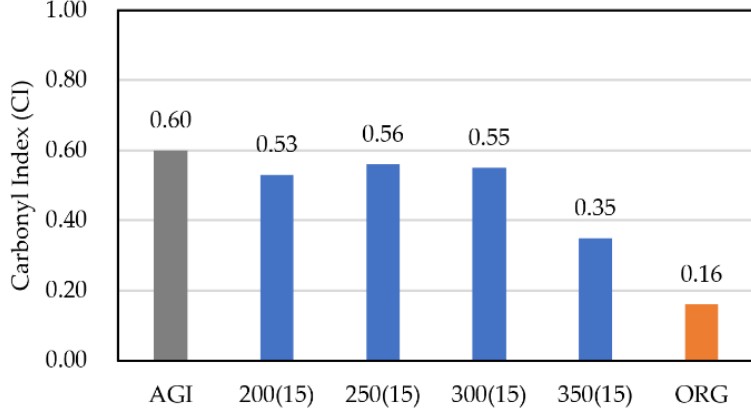

**Figure 17.** Carbonyl index (200–350 °C, 15 min).

Figure 18 shows the changes in the component ratio depending on the reaction temperature. The AGI increased the asphaltene and resin content compared to the ORG and significantly decreased the aromatic content. Conversely, the binder rejuvenated by hydrothermal decomposition tended to decrease the asphaltene and resin content and increase the aromatic content as the reaction temperature increased. The sample with a reaction temperature of 350 °C had a remarkable rejuvenating effect on the molecular weight distribution, CI, and component ratio.

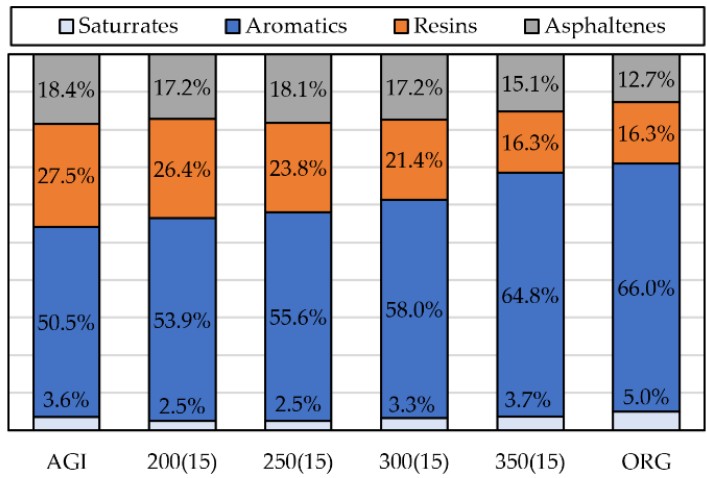

**Figure 18.** Composition (200–350 °C, 15 min).

### 4.2.2. Physical Properties of Rejuvenated Binder

Figures 19–21 show the changes in penetration, softening point, and elongation contingent on the reaction temperature. Compared to the ORG, the AGI had lower needle penetration and elongation and a higher softening point. Conversely, the binder, rejuvenated by hydrothermal decomposition, tended to show a gradual increase in needle penetration at a reaction temperature of 200–300 °C and a significant increase at 350 °C. From these changes in physical properties, the rejuvenating effect of the asphalt binder using the hydrothermal decomposition method was acknowledged, and the needle penetration, softening point, and elongation of the binder rejuvenated at 350 °C satisfied the standard values of the virgin binder (penetration: 60–80 (1/10 mm), softening point: 44.0–52.0 °C, elongation at 15 °C: ≥ 100 cm).

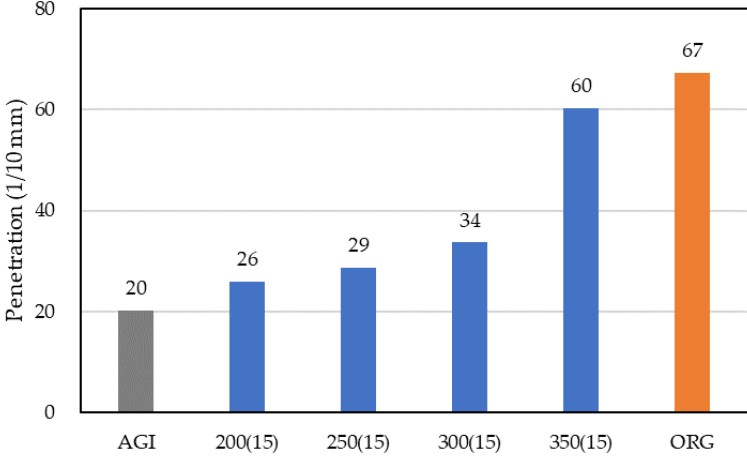

**Figure 19.** Penetration (200–350 °C, 15 min).

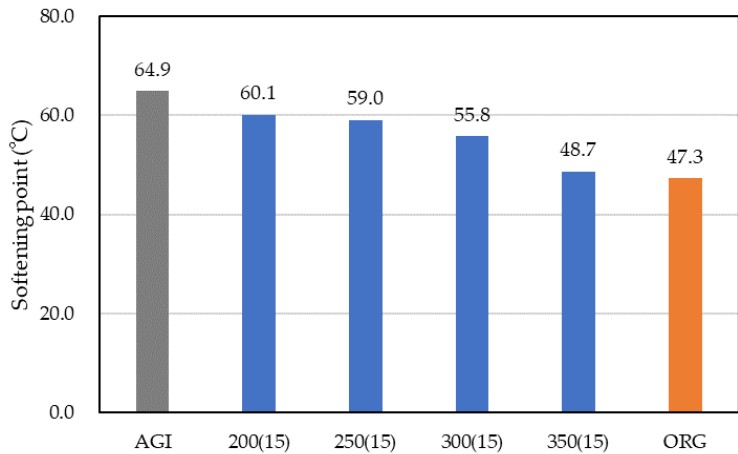

**Figure 20.** Softening point (200–350 °C, 15 min).

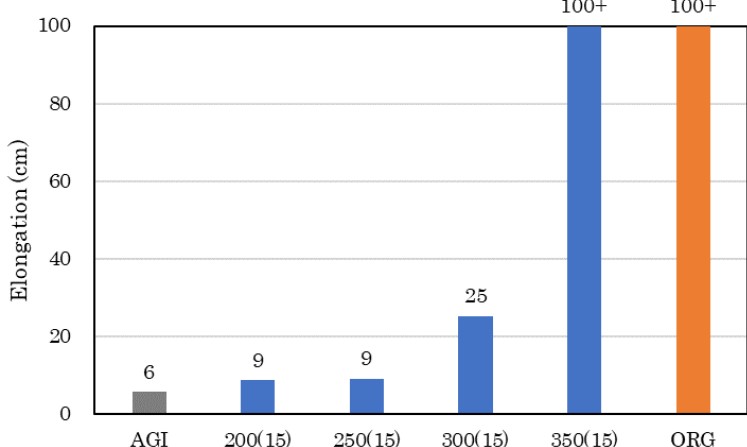

**Figure 21.** Elongation (200–350 °C, 15 min).

Based on these results, the rejuvenating effect of the binder by the hydrothermal decomposition method was dependent on the reaction temperature. The effect tends to improve significantly in the reaction temperature range 300–350 °C, and the relative permittivity, shown in Figure 7, is assumed to be involved. The study confirmed that a rejuvenating effect was obtained with a reaction time of 0–15 min, and that a binder rejuvenated through this method was less likely to have an increased asphaltene content due to aging.

## 5. Analysis

Based on these results, the quality of the aggregate recovered by the hot water rubbing method at 80 °C and 90 °C fully satisfied the standard values for virgin aggregate (density $2.45 \times 10^{-3}$ kg/m$^3$ or more, water absorption rate $\leq 3.0\%$, and wear and loss $\leq 30\%$) at all temperatures. Therefore, it is considered feasible that the aggregate separated from the binder and recovered by this method can be managed and used similarly as virgin aggregates.

In addition, to verify the applicability of this method to various binders, the separation and recovery of the aggregate using an RAP containing a modified binder as a specimen was attempted. An asphalt concrete block, in which the surface layer of the H-type modified binder and the base layer of the II-type modified binder, was excavated. As shown in Table 7, the quality of the aggregate recovered at separation temperatures of 80 °C and 90 °C did not change before and after binder extraction and satisfied the standard value of virgin aggregate [20].

**Table 7.** Properties of RAP containing modified binder.

| Properties | Unit | SR5–13m [1] | | R5–13m | SR1-5m | | | | R1–5m | | Virgin Aggregate (Standard Values) |
| --- | --- | --- | --- | --- | --- | --- | --- | --- | --- | --- | --- |
| | | 80 °C | 90 °C | | 80 °C | | 90 °C | | | | |
| | | | | | SR2.5–5m | SR1–2.5m | SR2.5–5m | SR1–2.5m | R2.5–5m | R1–2.5m | |
| Moisture content (After natural cooling) | % | 0.23 | 0.17 | - | 2.50 | | 1.97 | | - | | - |
| Old asphalt content | % | 0.87 | 0.70 | 2.95 | 0.07 | | 0.01 | | - | | - |
| Content of aggregates finer than 75 μm sieve | % | 0.1 | 0.1 | 2.3 | 0.2 | | 0.3 | | - | | - |
| Specific gravity | $\times 10^{-3}$ kg/m$^3$ | 2.63 | 2.66 | 2.67 | 2.67 | 2.67 | 2.66 | 2.66 | 2.67 | 2.65 | 2.45 or more |
| Water absorption | % | 0.42 | 0.50 | 0.79 | 0.50 | 0.82 | 0.55 | 0.83 | 0.79 | 1.00 | $\leq 3.0$ |
| Abrasion loss rate | % | 11 | 11 | 11 | 10 | - | 14 | - | 14 | - | $\leq 30$ |

[1] "m" is added to the end of the name of the sample containing a modified binder.

Conversely, fine particles (SR0–1) of ≤1 mm containing an aged binder were recovered by this method in the condition shown in Figure 22; therefore, their effective utilization must be examined.

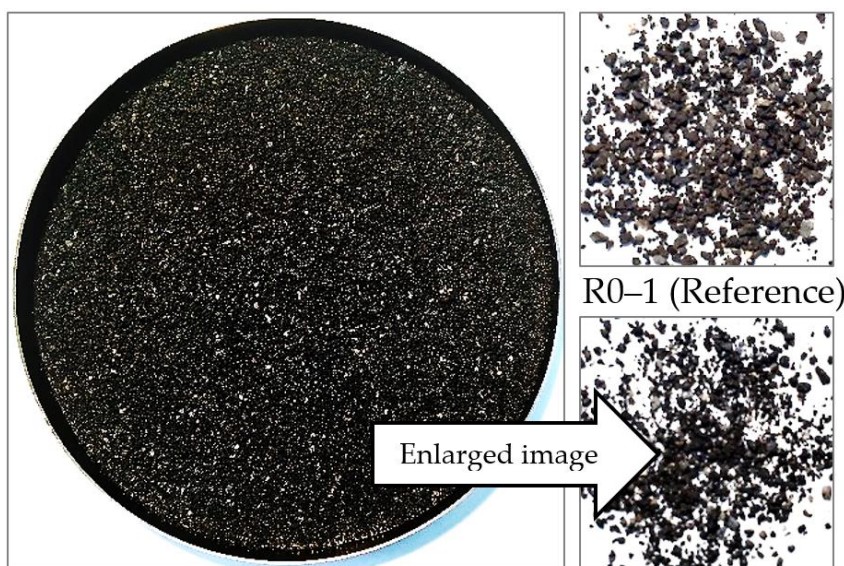

**Figure 22.** Appearance of SR0–1.

## 6. Conclusions and Recommendations

The results of this study confirmed that aggregates 1 mm and larger recovered from RAP by hydrothermal rubbing were classified as 5–13 mm and 1–5 mm with a significant reduction in the old asphalt binder film and were of similar quality to new aggregates. Hydrothermal decomposition of the accelerated aging of asphalt binder recovered both chemical and physical properties that were much closer to those of new asphalt. However, both methods must to be practical and efficient when upscaled for commercial applications. Because both methods use tap water and heat (70–90 °C and 300–350 °C) as resources, it will be important to select a heat source with a low environmental load and thermal recycling in the plant. It has been confirmed that the water used in these methods can be recycled and can meet drainage standards; however, the effect of tap water quality on the rejuvenating effects has not been clarified. In addition, it is necessary to determine the adaptability of water and heat to various RAPs, which includes binders that have been repeatedly rejuvenated. These findings should contribute to the establishment of sustainable recycling technologies for RAP.

**Author Contributions:** Conceptualization and methodology, K.A., Y.K. and S.A.; software, validation, formal analysis, data curation, writing original draft preparation, writing review and editing and visualization, K.A. and Y.K.; investigation, K.A.; resources, supervision, project administration and funding acquisition, Y.K. and S.A.; All authors have read and agreed to the published version of the manuscript.

**Funding:** This work was supported by JSPS KAKENHI Grant Numbers 23760411, 26820179, 19H02219.

**Data Availability Statement:** The data presented in this study are openly available in [repository name e.g., FigShare] at [doi], reference number.

**Acknowledgments:** The authors wish to thank KAKENHI (23760411, 26820179, 19H02219), Taisei Rotec Corporation, and CHIZAKIDORO Co., Ltd. for their support of this research. Also, the authors would like to acknowledge Hiroyuki Nitta and Yoko Kawashima of the PUBLIC WORKS RESEARCH INSTITUTE for their assistance in conducting the chemical analysis.

**Conflicts of Interest:** The authors declare no known competing financial interests or personal relationships that influenced the work reported in this study. The funders had no role in the design of the study; in the collection, analyses, or interpretation of data; in the writing of the manuscript, or in the decision to publish the results.

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
