# Peer review of "Technical Approaches to the Recycling of Reclaimed Asphalt Pavement into Aggregate and Binder"

_constrmater, doi:10.3390/constrmater2020007_

Round 1

Reviewer 1 Report

This study evaluated the technical approaches for recycling reclaimed asphalt pavement (RAP). In general, this is a very interesting research with well-designed experiments. However, there are some problems which the authors should look into.

  1. The literature review is not sufficient. The authors tried to focus on the separation of old asphalt and aggregate. However, in many countries such as America, people just directly use the RAP as aggregate and blend it with new asphalt. Therefore, the authors may explain the advantages and disadvantages of the two recycling methods. Why the separation of old asphalt and aggregate is needed?
  2. The moisture-induced stripping in asphalt pavement is a big problem since the water tends to displace the asphalt from aggregate, which is a thermodynamically favorable process. ("Measuring Moisture Damage of Asphalt Mixtures: The Development of a New Modified Boiling Test Based on Color Image Processing. Measurement, p.110699."). I believe the hot water rubbing method also takes advantage of the moisture damage, and the authors may explain this method in this way.
  3. The conclusion is too simple. Please give more description of the experimental results.

Author Response

2022/3/12

Dear Reviewer

Thank you very much for reviewing our manuscript and offering valuable advice. We have addressed your comments with pint-by-point response, and revised the manuscript accordingly.

Sincerely, Kengo Akatsu

Reviewer 2 Report

The manuscript needs to be edited by an English editor. There are, illogical sentences, unusual word choices that are not typically used for asphalt materials, and grammatical issues. The following identified examples are from the first 3 pages. Note that many other examples were found in the remainder of the manuscript, but were not listed below.   

Page 1, rows 30-31: Difficult to under

Page 2, rows 49-52: “treatment” is poor word choice and not understood. “condition” is more appropriate.

Page 2, rows 66-68: All of the text descriptions on Figure 3 are very poor and thus not understandable.

Page 6 and Page 7, Tables 2-4: Tables include a row titled “Particle quantity aggregate” What is this, why is not explained in the text?

Page 8: Figure 9: “SR%-13 After abstraction” what is abstraction? Is this a typographic error or word or ?

Page 9: Table 5: the title needs English editing.

Page 9: Table 5: the abrasion loss of the aggregates used are very low. Wy weren’t aggregates with higher abrasion loss included in the study? A significant amount of the aggregates used in the USA have abrasion loss significantly higher than 10-14%. In fact, 30-40% is common for Limestones. This brings into question the validity of the described process for “typical” aggregates used in the USA.

Page 10: Figure 13: an acronym used in the figure title is not defined.

Page 10, pages 235-238: This sentence is extremely difficult to interpret, not sure what is being stated?

Page 11, rows 260-263: Cannot understand this sentence. It is not clear what the author is trying to convey or what AGI is?        

Page 12, Table 7 and page 11, row 261: What is AGI, is has not been defined?

Page 12, Table 7: the penetration of the AGI is far less than the ORG, so the AGI material is much stiffer, but the term “rejuvenated” is be used to describe the outcome of the Hydropyrolysis method? That does not make sense. I give up on the binder analysis because it is not clear what the materials are or how they were treated. The is interesting but extremely frustrating to try to read and understand.

The manuscript appears to describe a set of tests on a single RAP source. And the abrasion loss of the aggregate was extremely low. 

The experimental design should have included multiple RAP sources from different aggregate sources, types, geology/minerology and multiple asphalt binder sources.  

Author Response

(The authors gave the same response as above.)

Round 2

Reviewer 1 Report

This paper has been revised based on the comments and should be ready for publication.

Author Response

Thank you very much for reviewing our manuscript and offering valuable advice. Revisions to the text will be made with input from the English editor review.

Reviewer 2 Report

This is interesting work. The revisions have improved the manuscript although there are still awkward sentences and use of terms that indicate an English editor review would improve the manuscript.

Author Response

(The authors gave the same response as above.)
